# Mental health and mental health help-seeking behaviors among first-generation voluntary African migrants: A systematic review

Edith N. Botchway-Commey[1,2]⊛*, Obed Adonteng-Kissi[3]⊛, Nnaemeka Meribe[4], David Chisanga[5], Ahmed A. Moustafa[6,7], Agness Tembo[8], Frank Darkwa Baffour[9], Kathomi Gatwiri[10], Aunty Kerrie Doyle[11], Lillian Mwanri[12], Uchechukwu Levi Osuagwu[11,13]*

1 Murdoch Children's Research Institute, Brain and Mind Group, Clinical Sciences, Melbourne, Victoria, Australia, 2 Department of Pediatrics, University of Melbourne, Melbourne, Victoria, Australia, 3 School of Arts and Humanities, Edith Cowan University, Southwest Campus, Bunbury, Australia, 4 Department of Politics, Media and Philosophy, La Trobe University, Melbourne, Victoria, Australia, 5 Department of Energy Environment and Climate Action, Agriculture Victoria, La Trobe University, Melbourne, Victoria, Australia, 6 Faculty of Society and Design, School of Psychology, Bond University, Gold Coast, Queensland, Australia, 7 The Faculty of Health Sciences, Department of Human Anatomy and Physiology, University of Johannesburg, Johannesburg, South Africa, 8 Faculty of Medicine and Health, Susan Wakil School of Nursing and Midwifery, The University of Sydney, Sydney, New South Wales, Australia, 9 School of Humanities, Social Science and Creative Industries, The University of Newcastle, Newcastle, New South Wales, Australia, 10 Faculty of Health, Centre for Children & Young People, Southern Cross University, Gold Coast, Queensland, Australia, 11 Translational Health Research Institute, School of Medicine, Aboriginal Health and Wellbeing CAG, Western Sydney University, Campbelltown, New South Wales, Australia, 12 Centre for Public Health Research Equity and Human Flourishing, Torrens University Australia, Adelaide, South Australia, Australia, 13 Bathurst Rural Clinical School, School of Medicine, Western Sydney University, Bathurst, New South Wales, Australia

⊛ These authors contributed equally to this work.
* L.Osuagwu@westernsydney.edu.au (ULO); edith.botchway@mcri.edu.au (ENB-C)

## Abstract

### Purpose

Mental health challenges are highly prevalent in African migrants. However, understanding of mental health outcomes in first-generation voluntary African migrants is limited, despite the unique challenges faced by this migrant subgroup. This review aimed to synthesize the literature to understand the mental health challenges, help-seeking behavior, and the relationship between mental health and mental health help-seeking behavior in first-generation voluntary African migrants living outside Africa.

### Methods

Medline Complete, EMBASE, CINAHL Complete, and APA PsychINFO were searched for studies published between January 2012 to December 2023. Retrieved articles were processed, data from selected articles were extracted and synthesized to address the study aims, and included studies were evaluated for risk of bias.

### Results

Eight studies were included, including four quantitative and four qualitative studies, which focused on women with postnatal depression. Mental health challenges reported in the

**Data Availability Statement:** Data available by uploading the minimal anonymized dataset necessary to replicate this study as supporting

information: S1 File S1A Appendix to S3D Appendix.

**Funding:** The author(s) received no specific funding for this work. The APC was funded by the Research Support Programme Fellowship from The Translational Health Research Institute (THRI), School of Medicine, Western Sydney University, Campbelltown, NSW, Australia, for one of the corresponding authors (ULO).

**Competing interests:** The authors have declared that no competing interests exist.

quantitative studies were depression, interpersonal disorders, and work-related stress. Risk (e.g., neglect from health professionals and lack of social/spousal support) and protective (e.g., sensitivity of community services and faith) factors associated with mental health were identified. Barriers (e.g., cultural beliefs about mental health and racial discrimination) and facilitators (sensitizing African women about mental health) of mental health help-seeking behavior were also identified. No significant relationship was reported between mental health and mental health help-seeking behavior, and the risk of bias results indicated some methodological flaws in the studies.

## Conclusion

This review shows the dearth of research focusing on mental health and help-seeking behavior in this subgroup of African migrants. The findings highlight the importance of African migrants, especially mothers with newborns, examining cultural beliefs that may impact their mental health and willingness to seek help. Receiving countries should also strive to understand the needs of first-generation voluntary African migrants living abroad and offer mental health support that is patient-centered and culturally sensitive.

## Introduction

For decades, migration has influenced human societies and regularly featured in society's cultural, political and economic dynamics, with the United Nations Population Fund reporting that around 244 million individuals lived outside their home country in 2015 [1]. In the same year, the United Nations High Commissioner for Refugees reported that more than one million migrants fled their country of origin to Europe by boat due to war, violence, and political and economic destabilization in their home countries [2,3]. More recently, populations are starting to migrate due to the negative effects of climate change [4], the COVID-19 pandemic, and the upsurge in global socio-political insecurity, socioeconomic problems, and conflicts [5,6].

A 2022 report from the European Union Institute for Security Studies stated that since 2010, the number of African migrants residing outside their home country has nearly doubled, reaching nearly 41 million [6]. In 2020, about 21 million of those were residing in another African country, whereas 19.5 million Africans lived outside Africa, including 11 million (56.4%) in Europe, 5 million (25.6%) in Asia, and 3 million (15.4%) in North America. The report further estimated that by 2030, Africa would continue to be viewed as a continent of emigration, with 429,000 more Africans projected to relocate from Africa than the total number of immigrants it will receive or host from other regions around the globe. Despite this significant migration trend among African migrants, and the known impact of migration on mental health globally [7,8], mental health in African migrants remains poorly researched [9].

Compared to migrants from other parts of the world, African migrants have shown greater vulnerability to mental health challenges such as depression, posttraumatic stress disorder (PTSD), and anxiety [10] since they often experience significant stressors [11] that increase their risk to these problems [12–14]. These stressors include struggles with changing family structure and dynamics, different cultural expectations, inadequate support systems (e.g., inability to access usual sources of support from extended family), and affirming practices of African migrants (e.g., actions or practices that validate their experiences and identities), which collectively affect their mental health and wellbeing [12,15]. In migrant men of African

origin, risk of mental health challenges has been associated with strains associated with adjusting to new roles and responsibilities in the destination country [16]. Studies have also shown that migrant women of African origin usually display an elevated risk of internalizing difficulties (e.g., depression) compared to men and to women of non-migrant/non-African backgrounds [17,18] due to migration-related (e.g., navigating health care system new country) and acculturation strains [19,20] and the increased risk of intimate partner violence [21] experienced by these women [16].

It has also been shown that migrant women are twice as likely to experience postnatal depression (PND) compared to non-migrant women due to factors such as low household income, low level of education, single parenting, migrating for marriage, and limited partner support [18,22]. For instance, a study of primiparous women in Australia, in which African women whose first language was not English were also studied, suggested that demoralization related to negative postnatal experiences and some of these connections are substantial even after considering demographics and other depressing issues [18]. Due to the higher risk of mental health issues among this migrant group, it is essential to create and apply mental health interventions that are culturally suitable and target stressors related to migration in this population.

There are various categories of migrant groups (e.g., first-generation vs second- generation migrants and voluntary vs non-voluntary migrants) but most studies on mental health in African migrants fail to acknowledge these subgroups [13,22]. First-generation migrants often migrate as adults and may be either non-voluntary migrants (i.e., people forcedly displaced from their home country due conflict or safety such as refugees and asylum seekers) or voluntary migrants (i.e., people who willingly migrate from their home countries for education or in search of greener pastures, including labor migrants, migrants moving to join their spouses etc.) [10]. Migrants classified as second/later-generation migrants are born in their host country to one or two parents born overseas [10]. Considering the profound differences in circumstances leading to migration especially between first-generation voluntary and non-voluntary migrants, it is essential to investigate the unique mental health challenges and associated factors in each migrant group. Such investigations are also essential due to the demonstrated role of ethnic identity in the wellbeing of migrants, where compared to second-generation migrants, first-generation migrants are at a greater risk of being affected by the source culture [23] and hence more vulnerable the stresses associated with change in culture.

To-date however, most studies on mental health in African migrants either lack this distinction or focus on first-generation non-voluntary migrants, such as refuges and asylum seekers [15,24–26]. This has led to a significant knowledge gap on mental health outcomes and help-seeking behavior in first-generation voluntary African migrants. Hence, there is currently no systematic review on studies assessing mental health outcomes and help-seeking behavior in first-generation voluntary African migrants. Systematically reviewing this literature will provide the much-needed information required to understand their mental health and help-seeking behavior and help develop tailored interventions to improve these outcomes. Focusing such a study on adults instead of minors can help identify the unique mental health challenges associated with the care-related responsibilities of migrating.

This systematic review will address this gap by synthesizing the literature on mental health and mental health help-seeking behavior of first-generation voluntary African adult migrants living outside Africa, by answering these three research questions: (1) What are the mental health challenges and associated risk and protective factors reported in first-generation voluntary African migrants?, (2) What are the barriers and facilitators of mental health help-seeking behaviors in this group of migrants?, and (3) What is the relationship between mental health and help-seeking behavior in first-generation voluntary African migrants?

## Materials and methods

This systematic review was conducted in accordance with the Preferred Reporting Items for Systematic Reviews and Meta-Analysis (PRISMA) guidelines [27]. The review was registered on the International Prospective Register of Systematic Reviews (PROSPERO) database, with a recorded registration number CRD42022340337.

### Search criteria

A comprehensive search strategy was developed to identify articles reporting on mental health outcomes and/or mental health help-seeking behaviors among African migrants living outside Africa. An assortment of terms related to these main concepts were used: ["Mental health" AND "mental health help-seeking behavior"], AND [African migrants OR first-generation African migrants OR a list of all African countries included (e.g., Ghana)], AND [Adults] (See detailed search criteria in the S1 File). Database searches were conducted in Medline Complete, EMBASE, CINAHL Complete, and APA PsychINFO for studies published between January 2012 to December 2023. A log of the search terms (S1 File) and search outcomes for each database is presented in S1A Appendix to S3D Appendix.

### Inclusion and exclusion criteria

Studies were included if they: (1) included first generation voluntary African migrants (including people migrating for education, labor migrants, migrants moving to join their spouses etc.), (2) evaluated mental health and/or mental health help-seeking, (3) included African migrants living outside the continent of Africa, since the commonalities among African cultures can significantly distinguish the migration experiences of Africans within Africa from those of African migrants residing outside the continent, (4) included adult participants (i.e., 18+ years old), (5) used a qualitative, quantitative, cross-sectional, or longitudinal design and had an abstract and full text available, (6) were published between January 2012 and December 2023 (since an in-depth analysis of mental health trends and treatment from 1997 to 2017 underscored the significance of considering historical context by demonstrating that health behaviors reported in 1990s data were shaped by different cultural conditions compared to data collected in the 2010s [28]), and (7) were peer-reviewed and published in the English language.

We excluded studies that: (1) did not include first-generation voluntary African migrants, (2) homogenized the data for non-voluntary migrants (i.e., refugees and asylum seekers) with that of voluntary migrants or first-generation with that of later generation migrants, (3) did not present separate outcomes for adults and children, or (4) did not contain original study data (e.g., book chapters, systematic reviews).

### Manuscript selection process

Articles identified from the databases were processed in COVIDENCE [29]. After all duplicates were detected and removed in COVIDENCE, two authors (OA-K and NM) independently screened the titles and abstracts for each article, EB-C and NM reviewed the full text of each article and selected the papers considered relevant for this systematic review. For any discrepancies in the paper selection, the decisions were discussed with EBC or ULO. Data extraction and synthesis

Data relevant to this systematic review were extracted from all selected papers and synthesized in Microsoft Excel, following the Strengthening the Reporting of Observational Studies in Epidemiology guidelines for reporting observational studies [30]. Relevant information

from papers extracted to populate these columns of the excel sheet: Authors, year country, study design, participant age, inclusion criteria, assessment methods, participant groups, primary outcomes, other outcomes, analysis methods, findings, study conclusions, strengths, and limitations. For the quantitative studies, a summary of the relevant results showing statistically significant effects (e.g., $p < 0.05$) are presented. Narrative summaries on perceptions around mental health problems or mental health help-seeking behavior are presented for the qualitative studies. Two authors extracted the data for this review (EBC and DC) and EBC drafted the results.

### Risk of bias assessment

The methodological quality of quantitative studies were assessed using the Quality In Prognosis Studies (QUIPS) tool [31], which assesses study quality across six domains: participation, attrition, prognostic factor measurement, outcome measurement, study confounders, and statistical analysis and reporting. Each risk of bias (RoB) domain included three to seven prompting items which were scored as Yes, No, Partial, or Unsure. Based on these item scores, each domain was determined to have either a low, moderate, or high RoB. In instances where an item/domain did not apply to a study (e.g., attrition in a cross-sectional study), the item/domain was scored as not applicable (N/A). An overall RoB rating was assigned to each paper based on the method proposed by Grooten and colleagues [32] as follows: Low RoB, if all domains were classified as having low RoB, or up to one moderate RoB; High RoB, if one or more domains were classified as having high RoB, or ≥3 were classified as having moderate RoB; Moderate RoB, all papers in-between were classified as having moderate RoB.

RoB for qualitative studies was evaluated using the Critical Appraisal of Qualitative Studies Worksheet from the Oxford Centre for Evidence-Based Medicine [33]. This tool evaluates the quality of studies using these ten items; whether the paper has a clear rationale, appropriateness of qualitative approach, sampling strategy, data collection method, data analysis and checks, description of researcher's position, what are the results, whether the results make sense, justifiable conclusions, and transferability of the findings to other clinical settings. Each item was rated as, Yes, No, or Unclear, and the overall RoB of each study was determined based on the number of criteria/items fulfilled on the checklist [34].

To ensure studies were evaluated appropriately, experts in quantitative (EBC and ULO) and qualitative (O.A-K and NM) research assessed the RoB for these respective study types and conflicts were resolved in meetings involving all four assessors and two other authors (L. M and D.C).

## Results

### Search results

A total of 9738 articles were generated from the four databases (Medline Complete = 3176, Embase = 255, APA PsychInfo = 3441, and CINAHL Complete = 2853), and 13 additional articles were identified from the reference lists of selected papers. Forty-five articles were included for full-text review, and eight were included in this systematic review. See the PRISMA flowchart in Fig 1.

### Study characteristics

The eight selected articles included four quantitative [35–38] and four qualitative [39–42] studies. The studies included first-generation voluntary African migrant samples from Canada, United Kingdom (UK), United States of America (USA), Spain, China, and Italy. Some studies

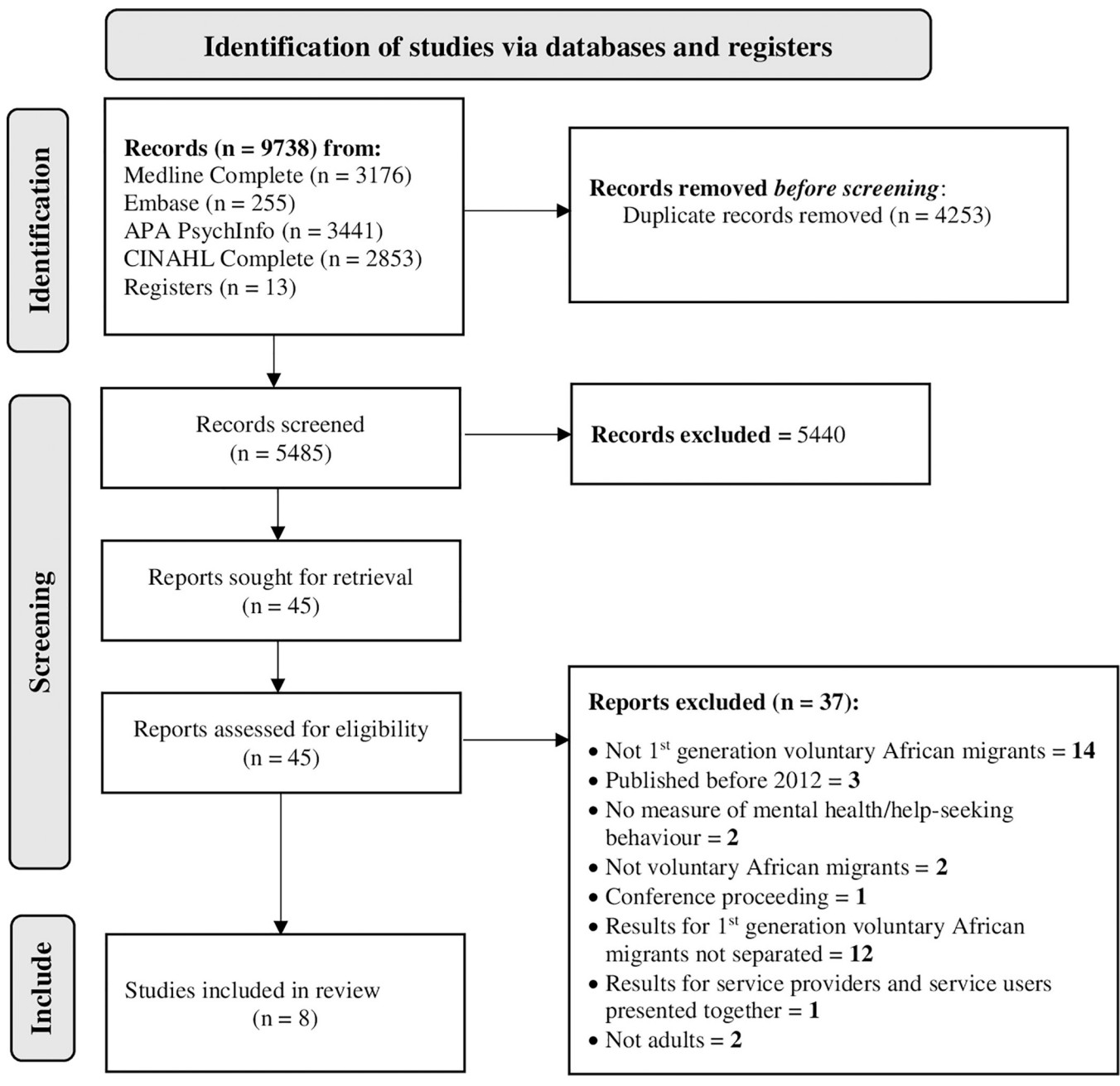

**Fig 1. PRISMA flowchart of included and excluded studies in a systematic review.**

noted the migrating country of their samples, with a limited number of African countries represented across studies: Ghana, Nigeria, Morocco, Zimbabwe [36,38,40–42]; with three studies stating that their sample were from sub-Saharan African [35,38,39]. Both quantitative and qualitative studies were cross-sectional, and used surveys and interview data collection methods, respectively. The quantitative studies included large sample sizes (i.e., 250 and 928), while the qualitative studies included 6 to 25 participants. All studies involved adults between 18 to 77 years old. Regarding sex representation, the quantitative studies included both men and women, although the proportion of female representation in the study by Capasso et al. [36]

Table 1. Study characteristics.

| Authors, Year, Country | Sample size (N) | Age (years) | Sex (%) | Sample Description | Study Design | Outcomes & Measures |
|---|---|---|---|---|---|---|
| Orjiako & So. (2014)[35], USA | 669 | 18–77 Mean = 34.1 | M: 57.3% F: 42.8% | African immigrants admitted to Lawful Permanent Residence Programs in the USA. | Cross-sectional study. Analyzed archival data from the US New Immigrant Survey (NIS) | **Mental health outcomes.** Depression was assessed with a study-designed questionnaire, which was found to be comparable (for face validity) to the symptoms of major depression listed in the DSM-IV-TR. **Help-seeking behavior.** Help-seeking behavior was assessed with a study-designed questionnaire, including 12 domains that evaluated the number of support systems used in the past 12 months. |
| Capasso et al. (2018)[36], Italy | 250 | Mean = 40.8 (SD = 3.51) | M = 90% F = 10% | Moroccan factory workers in Southern Italy | Cross-sectional study | **Mental health outcomes.** Interpersonal disorders, anxious-depressive disorders, work-related stress, and general health were assessed using a questionnaire. |
| Paloma et al. (2014)[37], Spain | 633 | Mean = 31.9 (SD = 8.5) | M = 48.2% F = 51.8% | Moroccan migrants living in Southern Spain, selected from 20 territorial units of Andalusia. | Cross-sectional study | **Mental health outcomes.** Wellbeing was assessed using the Satisfaction with Life Scale. |
| Yang et al. (2021)[38], China | 928 | Mean = 26 (SD = 8.7) | M = 62% F = 38% | sub-Saharan African migrants in China | Cross-sectional study | **Mental health outcomes.** Depression was assessed using the Centre for Epidemiologic Studies Depression Scale (CES–D) |
| Baiden & Evans (2021) [39], Canada | 10 | 25 to 40 | F: 100% | African newcomer women who birthed a baby in Canada within the past year. | Qualitative study, using open-ended semi-structured interviews | **Mental health outcomes.** Perceptions of mental health and mental health service utilization was assessed using one-on-one semi-structured interviews. |
| Gardner et al. (2014)[40], UK | 6 | 22 to 36 | F: 100% | West African mothers (Nigerians = 3; Ghanaians = 3), experiencing low mood in the postnatal period and living in Northwest of England. | Qualitative study, using semi-structured interviews. | **Mental health outcomes.** Lived experience of PND was explored using semi-structured interviews. |
| Ling et al. (2023)[41], UK | 6 | 18 to 55 | F: 100% | Nigerian migrant mothers in the UK who experienced PND | Qualitative study, using semi-structured interviews | **Mental health outcomes.** Semi-structured interviews were used to explore these women's lived experience of PND, their coping behaviors, and treatment experiences. |
| Dei-Anane et al. (2018) [42], UK | 25 | 18 to 45 | F: 100% | Ghanaian migrant women who had given birth in London. | Qualitative study, using semi-structured interviews. | **Mental health outcomes.** Semi-structured interviews were used to explore these women's perceptions about PND. |

**Abbreviations**: DSM-IV-TR, Diagnostic and Statistical Manual of Mental Disorders, 4th Edition, Text Revision; F: Female; M, Male; NIS, New Immigrant Survey; PND, Postnatal; Depression; SD, Standard Deviation; UK, United Kingdom; USA, United States of America.

was very low (10%). All the qualitative studies included only women in line with their focus on PND following childbirth. The characteristics of each paper are presented in Table 1.

## Mental health challenges and associated factors

The findings from studies on PND are distinguished from those examining other mental health issues in first-generation voluntary African migrants. Results are presented in Table 2.

**All other mental health challenges reported in first-generation voluntary African migrants.** Only one study reported on the prevalence rate of mental health problems.

**Table 2. Mental health and mental health help-seeking behavior in first-generation voluntary African migrant.**

| Authors, Year, Country | Study findings | Strengths | Limitations |
|---|---|---|---|
| **Orjiako & So. (2014)[35], USA** | **Mental health outcomes.** Depressive symptoms were associated with poor proficiency in the English language ($p$ = .026), but not with educational level, years of education in the USA, or years away from home country (all $p$ > .0.5). **Help-seeking behavior.** Help-seeking behavior was positivity associated with English proficiency ($p$ = .010) and level of education ($p$ = .002), but not with years of education in the USA, or years away from home country (both $p$ > 0.05). **Relationship between mental health and help-seeking behavior**: No significant relationship was identified between depressive symptoms and help-seeking behavior (support systems, religious variables), all $p$ > 0.05. | Investigated both mental health and mental health help-seeking behavior in African migrants. Included a large sample size. Made good efforts to statistically control for potential confounds (e.g., checked potential effect of age, sex and birth country on results). | Only one question was used to estimate the duration of residence in the USA, which does not provide insight into people's previous international immigration experience. Limited use of validated questionnaires. No Control group. |
| **Capasso et al. (2018) [36], Italy** | **Mental health outcomes.** Interpersonal/relational disorders: Workers with Type A behavior (CI = 0.255–0.749, $p$ < 0.05) and those who perceived high levels of rewards (CI = 0.210–0.621, $p$ < 0.05) were less likely to report relational disorders. Greater levels of inter-personal disorders were reported in Moroccan workers highly reliant on a strategy involving the search for identity and adoption of the host culture (CI = 1.043–3.058, $p$ < 0.05) and those experiencing racial discrimination (CI = 1.152–4.130, $p$ < 0.05). Anxious-depressive disorder: Perceiving high levels of rewards and having high levels of social inhibition (CI = 0.192–0.613, $p$ < 0.05) were associated with lower levels of anxious-depressive disorders, while high level of negative affect (CI = 1.115–3.448, $p$ < 0.05) was associated with higher levels of anxious–depressive disorders. Work-related stress: High perception of work-related stress was associated with Type A behaviors (CI = 1.235–2.761), favoring an affirmation/maintenance culture strategy (CI = 1.276–4.862) and having high perceptions about work demands (CI = 1.119–2.532). Those with objective coping mechanisms had lower perception of work-related stress (CI = 0.379–0.815). | Unique for assessing the application of the ethnicity and work-related stress model in African migrant workers, showing how individual, ethnic, and work characteristic influence mental health outcomes. Included a large sample size of Moroccan factory workers. | Lack of a control group, sample included a larger proportion of males compared to females, and limited control over some potential confounds (e.g., age, and sex) reduces robustness of results. Focus on a single ethnic origin limits generalizability to other African groups. The cross-sectional design precludes any reference to temporal or causal directions of observed statistical associations. There may be some biases related to inconsistencies in measures completed. For instance, the proportion of participants that completed measures in one vs two sessions was not specified. |
| **Paloma et al. (2014) [37], Spain** | **Mental health outcomes.** Wellbeing in Moroccan immigrants was positively associated with individual level factors (use of active coping strategies, $p$ = 0.002; satisfaction with the receiving context, $p$ < 0.001; and time of stay in Spain, $p$ = 0.001) and a contextual factor (cultural sensitivity of community services, $p$ = 0.004). | Included a large sample size. Used a robust method, including GIS and engagement of community Moroccan members to identify target population and to collect data. The analysis method was thorough, and results are clearly presented. | No control group included. Analysis did not control for potential confounds like age, marital status, and sex. Focus on a single ethnic origin limits generalizability to African migrant from other countries. The cross-sectional design precludes any reference to temporal or causal directions of observed statistical associations. |

*(Continued)*

**Table 2.** (Continued)

| Authors, Year, Country | Study findings | Strengths | Limitations |
|---|---|---|---|
| Yang et al. (2021) [38], China | **Mental health outcomes.** In the whole study sample, depression scores were greater in people with no fixed residence (95% CI: 0.2 to 5.9, $p < 0.05$), living in a rental apartment (2.4, 95% CI: 0.4 to 4.3, $p < 0.05$) compared to living in hotel, having unsatisfactory housing conditions (95% CI: 0.9 to 7.1, $p < 0.05$), and perceiving/experiencing negative attitudes from local people (95% CI: 3.0 to 12.2, $p < 0.05$). Depression was reported in 44% of the sample, based on the CES-D cut-off score of 16. In this subgroup of participants, depression was associated with lower satisfaction with housing conditions (OR = 1.7, 95% CI: 0.8 to 3.3, $p < 0.05$) and perceiving/experiencing negative attitudes from local people (OR = 4.5, 95% CI: 1.2 to 16.1, $p < 0.05$). | Large sample size. Controlled for several potential confounds in statistical analysis (e.g., age, sex, marital status, annual income, and recruitment means) | No control group. The cross-sectional design precludes any reference to temporal or causal directions of observed statistical associations. |
| Baiden & Evans (2021) [39], Canada | **Mental health outcomes**: Black African newcomer women described determinants of their a sense of mental sanity after birth to include mental strength, ability to nurture their child and home, their infant's state of wellbeing, ones willpower and faith in God, and spousal support. **Help-seeking behavior**: Participants preferred nonmedical treatments (e.g., spousal support and spirituality) over formal mental health services, some acknowledged the significance of the latter, and many advocated for a combined approach (medical + spiritual). Barriers to seeking mental health support services: cultural beliefs (e.g., bewitchment) and stigma around postnatal mental health challenges, racial discrimination from health providers during maternal care, temporary immigration status which often limits their accessibly to health care services, and stress of navigating the health system. Facilitators of mental health support services: Need to sensitize immigrant women about maternal mental health and postpartum mental health services, reach out to immigrant women, and provide services that protest their confidentiality (e.g., online services). | One of few studies to focus on Black African newcomer mothers who have not accessed mental health services. There is no evidence of familiarity or bias with the participant selection. | Small sample size. The study does not state the pedigree or prior experience of researcher(s) in qualitative research methods. Steps used in feminist ethnography were not described. |
| Gardner et al. (2014) [40], UK | **Mental health outcomes**: Five overarching themes emerged that described the experience of PND in these West African mothers: (1) conceptualizing postnatal depression, (2) isolation, (3) loss of identity, (4) issues of trust and (5) relationships as a protective factor. **Help-seeking behavior**: Participants stated that their cultural background made it difficult to disclose feelings of depression to people in their community. Women may want to present their emotional problems to health professionals (because conversations are confidential) instead of sharing with others within their community for fear of stigma. Cultural expectations of care are mismatched with women's new experiences. | First study to investigate the experiences of West African mothers with PND who live in the UK, and how they perceive and make sense of their experiences. The themes generated add to the body of existing research on PND in Black and ethnic minority populations and offer insight into the lived experience of West African women residing in England. Such insights are vital to deliver effective, culturally sensitive care. Piloting the interview schedule ahead of administering is a strength of the study. | Involvement with services will inevitably have influenced participants' experiences. Interviewing only women who spoke English limits the generalizability of the results. Limited sample size. Other potential sources of bias include, not stating the pedigree of interviewer |

*(Continued)*

**Table 2.** (Continued)

| Authors, Year, Country | Study findings | Strengths | Limitations |
|---|---|---|---|
| Ling et al. (2023) [41], UK | **Mental health outcomes:** Three overarching themes and seven sub-themes emerged that described Nigerian mothers' experience of PND in the UK: (i) Socio-cultural factors (inter-generational expectation to be strong; cultural perceptions of shame and stigma; transitions/adjusting to a new culture); (ii) What about me? The neglected nurturer (experiences of treatment; pretending to be OK); and (iii) Loneliness and coping (lack of support from partner; self-reliance).<br>**Help-seeking behavior:**<br>Participants' efforts to reach out to professionals were met with unsatisfactory responses, including neglect from health visitors, midwives, and GPs; not feeling heard by GPs; and lack of support from spouse. | First study to focus on the experiences of first-generation Nigerian migrant mothers with PND in the UK, and the data provides valuable insights into the experiences of these women. | Interviewing only women who spoke English limits the generalizability of the results to all Nigerian mothers with PND in the UK. Limited sample size. Other potential sources of bias include, not stating the pedigree of interviewer, and not piloting semi-structured interview guide. |
| Dei-Anane et al. (2018) [42], UK | **Mental health outcomes:** Worry or sadness during the postnatal period was attributed to breastfeeding, infant's temperament, lack of support from partner, and accommodation.<br>**Help-seeking behavior:**<br>Most women did not seek help from professionals for their mental health problems because they had negative experiences/felt neglected by professionals. Coping strategies used in dealing with stress during the postnatal period included, brief period of help from family and friends in the UK, relying on religious leaders, family, and relatives in Ghana for emotional support, and remaining positive (since PND is perceived as part of their motherhood experience). | First study to focus on the experiences of first-generation Ghanaian migrant mothers with PND in the UK, and the data provides valuable insights into the experiences of these women, exploring how their cultural background uniquely influences their experiences of PND. | Limited sample size.<br>Other potential sources of bias include, not stating the pedigree of interviewer, not piloting semi-structured interview guide, not providing information about language in which interviews were conducted and whether some participants needed translation, and not stating who conducted interviews/led data analysis. |

**Abbreviations**: CI, Confidence Interval; PND, Postnatal Depression; UK, United Kingdom; USA, United States of America.

Specifically, Yang et al. [38] investigated rate of depression in sub-Saharan African migrants in China and reported a prevalence rate of 44% based on a CES-D cut-off score of 16. Depression in this subgroup of migrants was associated with lower satisfaction with housing conditions and perceiving/experiencing negative attitudes from local people. In addition to these factors, having no fixed residence and living in a rental apartment were significantly associated with greater risk of depression in the whole sample.

The three other quantitative studies [35–37] assessed the factors associated with mental health outcomes in their respective samples. Orjiako and So [35] investigated acculturation stress factors that predict depressive symptoms in a sub-Saharan African sample of 669 adults admitted to Lawful Permanent Residence Programs in the USA. Their results showed a strong association between proficiency in the English language and depression symptoms, while depression was not associated with factors like years of education or number of years away from home country. Capasso et al. [36] tested associations between individual, ethnic, and work characteristics and psychophysical health outcomes, using an ethnicity and work-related stress model in Moroccan factory workers in Southern Italy. Results showed a lower risk of interpersonal disorders in people with Type A behavior trait (i.e., where individuals show a strong sense of time urgency and competitiveness) and people who perceived high levels of

rewards, while those with high need to identify/adopt the host culture and those experiencing racial discrimination were at a greater risk of interpersonal disorders. Anxious-depressive symptoms in that study were associated with these risk factors (i.e., high level of negative affect) and protective factors (i.e., perceiving high levels of rewards and having high levels of social inhibition) factors. Likewise, high perceptions of work-related stress were associated with type A behaviors, favoring an affirmation/maintenance culture strategy, and having high perceptions about work demands, while favoring objective coping strategies was associated with lower perceptions of work-related stress. It should be noted that the majority of their sample were male. In a related study, Paloma et al. [37] assessed the factors associated with wellbeing in 633 Moroccan immigrants in Southern Spain, with the aim of developing a predictive model of the wellbeing of this population. Wellbeing in Moroccan immigrants was positively associated with individual level factors such as use of active coping strategies (i.e., believing that social change is possible and believing in one's ability to influence their context); satisfaction with the receiving context (i.e., to be satisfied with one's neighborhood and living situation); and time of stay in Spain. Cultural sensitivity of community services (i.e., the degree of cultural sensitivity of health services, police, social services, and public administration in communities where migrants live) was also associated with better wellbeing in these migrants.

**PND in first-generation voluntary African migrants.** All four qualitative studies investigated PND in first-generation voluntary African migrants [39–42], with three based on women in the UK [40–42]. Baiden & Evans [39] explored sociocultural factors that impact Black African newcomer women's perception of mental health and the use of mental health services within a year after childbirth in Canada. Findings showed that these women perceived that their mental wellbeing after birth is influenced by their mental and cultural resilience, willpower and faith in God, ability to nurture their child and home, their infant's state of wellbeing, and spousal support. Gardner at al. [40] explored the lived experience of PND in West African mothers living in the UK. Although they experienced symptoms of PND, these women referred to their experience as "social stress" or "stress". PND was attributed to lack of support in community, isolation, worry over family's financial status, and not having family nearby. In their view, PND resulted from social stress. In addition to conceptualization, four other dominant themes emerged on PND: (1) isolation–a lack of social, cultural support (i.e., lack of hands-on support for mother and child from immediate family and community), emotional support (i.e., having someone to talk to), and professional support (i.e., lack of on-going support from professional postpartum); (2) loss of identity, including loss of one's old self and old life (mourning the loss of who they once were); (3) issues of trust—distrust in others due to fear of betrayal and fear of stigma (from community members); (4) relationships facilitate recovery–work and other activities are helpful distractors, better relationship with baby was helpful for recovery, maternal support services and groups help reduce distress and facilitate recovery. In a related study, Ling et al. [41] assessed the lived experiences of Nigerian mothers in the UK who had experienced PND. Three main themes were identified, including seven sub-themes: (i) Socio-cultural factors (inter-generational expectation to conform to the strong Black woman identify; cultural perceptions of shame and stigma around PND made it difficult for women to open up; and transitions/adjusting to a new culture led to isolation and loss of community support); (ii) What about me? The neglected nurturer (neglect from health professionals; pretending to be OK to allow professionals to perform their duties); and (iii) Loneliness and coping (lack of emotional and practical support from partner often led to isolation, hopelessness, and suicidal ideation; self-reliance since participants were averse to the only help offered-medication therapy). In a similar study, Dei-Anane et al. [42] explored perceptions about PND in Ghanaian migrant women in London, and identified that PND in these women was attributed to breastfeeding (causing pain and discomfort, although breastfeeding was also

perceived as a sign of good motherhood), infant's temperament (poor sleeping, being demanding), lack of support from partner, and accommodation problems (limited housing space).

Taken together, the four quantitative studies provide insights into the risk and protective factors associated with mental health outcomes in first-generation voluntary migrant populations, while the four qualitative studies highlight the risk and protective factors associated with PND in first-generation voluntary African migrant women.

## Mental health help-seeking behaviors

Barries and facilitators of mental health help-seeking behavior were evaluated in one quantitative study [35] and three qualitative studies [39–41], and assessed using factors like number of support systems accessible to a person. Barriers were explored only in the studies, which investigated PND in migrant women of African origin [39–41]. Barriers identified included African cultural beliefs about mental health (e.g., bewitchment), cultural expectation to be a strong woman, cultural shame and stigma around mental health challenges, racial discrimination from health providers during maternal care, neglect from health visitors and midwives due to excessive focus on the child's health and limited focus on the mother's wellbeing, neglect from GPs, and GPs limited tendency to involve new mothers in treatment decisions, temporary immigration status (which often limits their accessibly to health care services), stress of navigating the health system [39,41,42], lack of support from partner, and lack of trust of others in community due to fear of stigma and profiling [40,41]. Facilitators of help-seeking behaviors reported in the other studies were proficiency in the English language ($p = 0.010$) and a higher level of education ($p = 0.002$)- in the USA migrant group [35]; sensitizing migrant women about maternal mental health and postpartum mental health services, reaching out to immigrant women, providing services that protest their confidentiality (e.g., online services) [39], and having access to health professionals that could provide a safe space for confidential conversations–in new-born mothers with PND [40]. See Table 2.

## Relationships between mental health and help-seeking behavior

The relationship between mental health and mental health help-seeking behavior was assessed in only one study [35]. Results showed no significant relationship between depression and help-seeking behaviors like number of support systems, number of religious studies attended in the USA, and number of religious affiliations (all $p > 0.05$) [35]. See Table 2.

**Table 3. Risk of bias results—quantitative studies.**

| Risk of bias item | Capasso et al. (2018) [36], Italy | Orjiako & So (2014) [35], USA | Paloma et al. (2014) [37], Spain | Yang et al. (2021) [38], China |
|---|---|---|---|---|
| Study Participation | Moderate | High | Low | Low |
| Study Attrition | N/A | N/A | N/A | N/A |
| Prognostic Factor Measurement | Low | Low | Low | Low |
| Outcome Measurement | Low | Low | Moderate | Low |
| Study Confounding | Moderate | Moderate | Moderate | Low |
| Statistical Analysis & Reporting | Low | High | Low | Low |
| **Overall Risk of Bias** | **Moderate** | **High** | **Moderate** | **Low** |

**Note**: Overall Risk of Bias (RoB) = based on the method proposed by Grooten and colleagues [32] as follows: Low RoB, if all domains were classified as having low RoB, or up to one moderate RoB; High RoB, if one or more domains were classified as having high RoB, or ≥3 were classified as having moderate RoB; Moderate RoB, all papers in-between were classified as having moderate RoB.

**Table 4. Risk of bias resutls—qualitative studies.**

| Risk of bias item | Gardner, et al. (2014) [40], UK | Baiden et al. (2021) [39], Canada | Ling et al. (2023) [41], UK | Dei-Anane et al. (2018) [42], UK |
|---|---|---|---|---|
| 1. Study rationale and question | Yes | Yes | Yes | Yes |
| 2. Appropriateness of qualitative approach | Yes | Yes | Yes | Yes |
| 3. Sampling strategy | Yes | Yes | Yes | Yes |
| 4. Data collection method | Yes | Yes | Yes | Yes |
| 5. Data analysis and check | Yes | Yes | Yes | Yes |
| 6. Description of researcher's position | No | No | Yes | No |
| 7. What are the results | Yes | Yes | Yes | Yes |
| 8. Whether the results make sense | Yes | Yes | Yes | Yes |
| 9. Justifiable conclusions | Yes | Yes | Yes | Yes |
| 10. Transferability of the findings to other clinical settings | No | No | No | No |
| **Overall Risk of Bias** | **8/10** | **8/10** | **9/10** | **8/10** |

**Note**: Overall Risk of Bias = Total number of domains scored as "Yes".

### Risk of bias

As presented in Table 3, one quantitative study was rated as having a low RoB [38] and the other three studies were rated as having a moderate and high RoB [35–37]. These later studies effectively met the quality standards for most RoB criteria, but they received lower ratings, primarily due to minimal efforts in controlling potential confounding factors.

Results of the RoB assessment for the qualitative studies [39–42] are presented in Table 4. All studies fulfilled 8 to 9 of the 10 RoB criteria, reflecting an acceptable quality [34]. Three of the studies failed to describe the position/qualifications of the qualitative researcher/interviewer, and partly owing to the subjective nature of qualitative studies, all studies were found to have limited transferability to other clinical settings.

## Discussion

This systematic review investigated the mental health challenges and associated factors, and the relationship between mental health and mental health help-seeking behavior in first-generation voluntary African migrants. Eight studies were included in this review [35–42], including four quantitative studies involving African migrants in the USA, Italy, Spain, and China and four qualitative studies on PND, involving African migrant women in Canada and the UK. The findings highlight the nature of mental health challenges experienced by these migrants (e.g., depression), the associated risk factors (e.g., isolation), and protective factors (e.g., access to cultural sensitivity of community services and faith). Some barriers (e.g., neglect from health professionals), and facilitators (e.g., level of education) of help-seeking behavior in this migrant group were also reported across studies. No significant relationship was reported between mental health and mental health help-seeking behavior. These findings provide preliminary insights about factors that can be targeted to improve mental health and mental health help-seeking behavior and highlight the need for more research on these topics in first-generation voluntary African migrants.

### Mental health challenges and associated factors

Concerning the mental health challenges and associated factors in first-generation voluntary African migrants, the studies involving samples other than women with PND showed that

these migrants experienced depression, anxious-depressive disorders, inter-personal disorders, and stress [35–38]. Only one study investigated the prevalence rate of these problems, reporting the 44% of sub-Saharan African migrants in China experienced depression [38]. This scarcity of data highlights the need for more studies investigating the prevalence of mental health challenges in this group of African migrants.

Protective and risk factors associated with mental health challenges in those studies were also identified [35–38]. Protective factors include proficiency in the English language, Type A personality trait, using active coping strategies, satisfaction with the receiving context, and cultural sensitivity of community services (e.g., including health care, police services) [35–37]. These factors mostly point to intrinsic traits of the migrant, but the latter highlights the importance of ensuring that service providers who attend to migrant communities are diverse, to some extent representative of the communities they assist, and use culturally informed and sensitive approaches to service delivery. The reported influence of proficiency in the language of the designation country on mental health, which was the English language in the context of Orjiako & So's study [35] resonates with previous studies showing similar results in migrant groups [43,44], and highlights the need for initiatives supporting language proficiency in African migrants. Type A personality trait was associated with lower risk of relational disorders, but higher risk of job-related stress in Moroccan factory workers, and some studies involving non-migrant populations (e.g., in Swedish and Indonesian students) [45,46] have also demonstrated a link between this personality trait and increased stress. This suggests that this personality trait could be helpful for integration and high achievement across cultures, but education around stress management, using healthy coping strategies can be beneficial for better outcomes in individuals with this trait, and especially in African migrants to limit their migration-related stress.

Risk factors included racial discrimination, a high need to fit into the host country's new culture, lower satisfaction with housing conditions, perceiving/experiencing negative attitudes from local people, having no fixed residence, and living in a rental apartment [36,38]. In a previous systematic review investigating psychological distress in migrant populations (including non-voluntary migrant samples), poor mental health was associated with traumatic events prior to migration, poorly planned/illegal migration, low level of acculturation, living alone in the host country, and perceived discrimination [47]. The notable distinctions in the factors they identified compared to those found in the current review highlight the unique set of risk factors influencing mental health in first-generation voluntary African migrants compared to migrant groups in general. This distinction may be because voluntary African migrants often have a different psychological profile compared to non-voluntary migrants (e.g., higher risk of trauma in voluntary migrants, often due to war in country of origin), and may have a different set of challenges (e.g., excelling in education or work) compared to the latter group.

The findings from the current review indicate that mental wellbeing in first-generation voluntary African migrants is influenced by a set of personal factors (e.g., sense of self-reliance, resilience, using active coping strategies) and system level factors (e.g., greater confidence in mental health service providers, safe housing, access to cultural informed and sensitive community services, and services that prioritize racial safety).

This review also included four studies exploring perceptions [39,42] and lived experience [40,41] of PND in first-generation voluntary African migrant women, all of whom emigrated from sub-Saharan Africa. Falah-Hassani et al. [22] reported that migrant women are twice as likely to experience PND compared to non-migrant women, highlighting a greater vulnerability in this group. Interestingly, women in two of the studies perceived PND in a manner distinct from Western perspectives, frequently opting to characterize it as "social stress", "mental stress" or stress [39,40]. This highlights a fundamental difference in conceptualization and the

need to understand this condition and the care needs of these women from a culturally sensitive perspective. Factors perceived to influence risk of PND across studies included isolation due to lack of social, cultural, emotional support and on-going professional support; loss of self and identity; lack of trust in community due to fear of stigma and betrayal [40,41]; ability to nurture their child and home; their infant's state of wellbeing; worry over family's financial status; lack of spousal support [39], inter-generational expectation to conform to the strong Black woman identify; cultural perceptions of shame and stigma around PND; transitions/adjusting to a new culture leading to isolation and loss of access to community; tendency to be self-reliant due to lack of support [41] breastfeeding (due to associated pain and discomfort); infant's temperament (e.g., sleep problems); and accommodation problems (e.g., limited space) [42].

While a few of these factors have been linked to PND in other women of non-Western backgrounds [48,49] most of the factors reflect self-reliance and are linked to the notion of presenting as a strong black woman, which may be a good defense against mental oppression, but is often linked with increased risk of mental health issues [50,51]. A systematic reviews of risk factors associated with PND in migrant women (including non-voluntary migrants), identified low household income and education, single parenting, migrating for marriage, limited partner support, and history of violence [22], while another involving women experiencing PND in Asian cultures identified factors such as antenatal depression, unwanted pregnancy, poverty, and preference for infants' gender [52]. Although some similarities exist between the factors identified in these previous reviews and the current one, our review linked PND with mostly cultural (e.g., believes and stigma about PND), relational (e.g., isolation and distrust), and systemic (e.g., lack of culturally sensitive support, neglect from health professionals) factors in voluntary African migrant women. Our findings highlight their need for trusting and supportive relationships and mental health professionals that could provide safe space for this group of migrant women. It could be helpful to invest in initiatives that support/promote existing mental health protective factors in this group of migrant women, including their resilience, spirituality, relationships (with work, with baby, maternal support services), and connection with family in their home country [40–42].

## Mental health help-seeking behaviors

This review presented interesting findings regarding the barriers and facilitators of mental health help-seeking behavior, with one study identifying English proficiency and level of education as facilitators in a group of in first-generation voluntary African migrants in the USA.

The studies investigating PND in women of African origin mostly reported that these women were often reluctant to seek help for their mental health challenges [39–41]. Barriers of help-seeking behavior reported in those studies included cultural beliefs about mental health (e.g., bewitchment); racial discrimination from health professionals; temporary immigration status [39]; neglect from health professionals, feeling less heard by health professionals, and lack of support from partner/spouse. Facilitators reported in those studies were sensitizing African migrant women about postnatal mental health; reaching out to African migrant women; prompting/questioning African women about their wellbeing after birth; and providing services that protect their confidentiality (e.g., online services) [35,39,41]. These findings indicate that a combination of personal, cultural, and systemic/environmental factors influenced mental health help-seeking behavior in first generation voluntary African migrants.

Mental health help-seeking behavior has been linked to several factors in other migrant groups: Polish migrants in the UK (barriers—older age, mental health stigma, knowledge of the health system; facilitators–education) [53]; Afghan and Iraqi refugees in Australia (barriers—not requiring interpreters and knowing how to navigate services. facilitators—older age, and

poor overall health) [54]. Despite the overlap between the factors identified in this review and previous studies, some key disparities exist (e.g., cultural believes about mental health, poor treatment experience with health professionals, and temporary immigration status in first generation voluntary African migrants) that reiterate previous calls for the need for targeted mental health promotions that are culturally attuned to the characteristics of specific groups [54]. The findings provide preliminary insights into factors influencing mental health help-seeking behavior, particularly in African migrant women with PND. They highlight the need for African migrants to review their own understandings of mental health and the internalized cultural messaging (e.g., shame and stigma associated with seeking help, and feeling weak for seeking help) that may prevent them from seeking mental health support. The findings also emphasize the need for health professionals, particularly GPs, midwives, and home visitors attending to these women to provide patient-centered, culturally sensitive, and more supportive (e.g., ask new mothers about their wellbeing) care to these women.

Interestingly, the only study investigating the relationship between mental health and mental health help-seeking behavior in this review did not find a significant link between these two outcomes. While this finding may be related to some of the methodological concerns of that study (e.g., using one question to estimate the length of immigration in the USA and limited use of validated questionnaires), it also highlights the need for more exploration of this question.

## Review strengths and limitations

This review has some limitations. (1) The number of studies included in this review is small, limiting the number of papers included in the data synthesized, and the robustness of the findings. (2) The quality of the data synthesis is limited by the fact that methodological approaches used in the included papers were very different (e.g., sample types differed across papers). (3) As indicated in the risk of bias results, the methodological rigor of the quantitative studies included in this review is limited (e.g., lack of control group) and may have affected the quality of the results presented. (4) The ability to apply these findings to all first-generation African migrants is limited because some studies did not mention the migrating country of their samples, and the ones that did seem to include a limited number of African countries. (i.e., Morocco, Ghana, Nigeria). (5) The generalizability of the findings to all first-generation African migrants is limited because the review does not include so many subgroups of these migrants: student groups, married vs. unmarried migrants, younger vs. older migrants). (6) The quality of data presented on the relationship between mental health and mental health help-seeking behavior is limited by the fact that only one study investigated this question.

Despite these limitations this review has several strengths, including (1) Being the first review to synthesize the data on mental health and help-seeking behavior in first-generation voluntary African migrants. (2) Including data from both quantitative and qualitative studies. (3) Providing preliminary insights into factors that impact the mental health in this group of migrants, that can be targeted to improve their mental health and wellbeing. (4) Using gold-standard systematic review methodology.

## Conclusions

This systematic review synthesized the literature on mental health and help-seeking behavior in first-generation voluntary African migrants. It provides preliminary insights into some of the mental health problems reported in this subgroup of African migrants, and the associated risk and protective, as well as the barriers and facilitators associated with mental health help-seeking behavior in this migrant group. These preliminary findings highlight the unique set of

factors affecting mental health and help-seeking behavior in this subgroup of African migrants, and show the dearth of research in this population; indicating the need for more rigorous studies on these topics. The findings provide valuable insights about the need for these African migrants, particularly mothers with newborns to reflect on cultural believes that hinder their mental health and help-seeking behavior, and for the broader African migrant community to reflect on practices that create a supportive space for its members. Receiving countries should also strive to understand the needs of first-generation voluntary African migrants and offer mental health support that is patient-centered and culturally sensitive.

## Supporting information

**S1 File. Detailed search criteria.**
(DOCX)

**S2 File. PRISMA checklist.**
(DOC)

**S1 Appendix.** **A**. CINAHL Search Strategy 15.07.2022. **B**. Embase Search Strategy 15.07.2022. **C**. Medline Complete Search Strategy 15.07.2022. **D.** PsychInfo Search Strategy 15.07.2022.
(ZIP)

**S2 Appendix.** **A.** CINAHL Search Strategy 23.05.2023. **B.** Embase Search Strategy 23.05.2023. **C.** Medline Complete Search Strategy 23.05.2023. **D.** PsychInfo Search Strategy 23.05.2023.
(ZIP)

**S3 Appendix.** **A.** CINAHL Search Strategy 16.12.2023. **B.** Embase Search Strategy 17.12.2023. **C.** Medline Complete Search Strategy 16.12.2023. **D.** PsychInfo Search Strategy 17.12.2023.
(ZIP)

## Author Contributions

**Conceptualization:** Edith N. Botchway-Commey, Obed Adonteng-Kissi, Nnaemeka Meribe, David Chisanga, Ahmed A. Moustafa, Agness Tembo, Frank Darkwa Baffour, Kathomi Gatwiri, Aunty Kerrie Doyle, Lillian Mwanri, Uchechukwu Levi Osuagwu.

**Data curation:** Edith N. Botchway-Commey, Obed Adonteng-Kissi, Nnaemeka Meribe, David Chisanga, Lillian Mwanri.

**Formal analysis:** Edith N. Botchway-Commey, Nnaemeka Meribe.

**Investigation:** Edith N. Botchway-Commey, Uchechukwu Levi Osuagwu.

**Methodology:** Edith N. Botchway-Commey.

**Project administration:** Uchechukwu Levi Osuagwu.

**Writing – original draft:** Edith N. Botchway-Commey, Obed Adonteng-Kissi, Nnaemeka Meribe, David Chisanga, Uchechukwu Levi Osuagwu.

**Writing – review & editing:** Edith N. Botchway-Commey, Obed Adonteng-Kissi, Nnaemeka Meribe, David Chisanga, Ahmed A. Moustafa, Agness Tembo, Frank Darkwa Baffour, Kathomi Gatwiri, Aunty Kerrie Doyle, Lillian Mwanri, Uchechukwu Levi Osuagwu.

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
