## [Decision Letter · Decision Letter 0]

25 Sep 2023

PONE-D-23-19022Mental health and mental health help-seeking behaviours among first-generation voluntary African migrants: A systematic reviewPLOS ONE

Dear Dr. Edith Nardu Botchway-Commey,

Thank you for submitting your manuscript to PLOS ONE. After careful consideration, we feel that it has merit but does not fully meet PLOS ONE’s publication criteria as it currently stands. Therefore, we invite you to submit a revised version of the manuscript that addresses the points raised during the review process. Please submit your revised manuscript by Nov 09 2023 11:59PM. If you will need more time than this to complete your revisions, please reply to this message or contact the journal office at plosone@plos.org. Please include the following items when submitting your revised manuscript:A rebuttal letter that responds to each point raised by the academic editor and reviewer(s). You should upload this letter as a separate file labeled 'Response to Reviewers'.A marked-up copy of your manuscript that highlights changes made to the original version. You should upload this as a separate file labeled 'Revised Manuscript with Track Changes'.An unmarked version of your revised paper without tracked changes. You should upload this as a separate file labeled 'Manuscript'.

We look forward to receiving your revised manuscript.

Kind regards,

Sharada P Wasti, Ph.D., MSc, MHCM, MA, PGDHCM

Academic Editor

PLOS ONE

Journal Requirements:

5. We are unable to open your Supporting Information file CINAHL Complete_15.07.2022.2022. Please kindly revise as necessary and re-upload.

Additional Editor Comments:

Thank you so much for submitting your manuscript to PLOS One. It was a pleasure reading and learning about your work and finding scholarly merit and practical relevance in this work. However, I believe it will benefit from improved quality and readability taking into consideration the following comments/concerns before accepting for publication in this journal. Here are the concerns to improve the manuscript:

• Abstract section para 44 either changes the objective to an introduction or makes a very clear objective and states only the objective not background information.

• Methods section add a sentence about how you synthesised data.

• Para 91 -92 reference number 14 add one more citation or correct the writing partners of para 91.

• Similar way para 93 studies have …. But only one reference is evident in para 96 ref number 96. Either add references or change the way of presentation of para 93-96.

• Insert a citation in para 102-112 to claim your statement.

• Citations are missing in para 115 – 123 to claim your arguments.

• Make consistency of your writing in headings and sub-headings first letter is capitalised and small i.e. 2.1 and 2.2 which are not consistent.

• Make a consistent presentation and correct para 142-143 and state January 2012 – May 2023.

• Make consistency para 146 including your data including month in 2012 which is missing and explain the reason why only chose 2012 which is not clear.

• Methods section there is no data synthesis section and not clear how you synthesise your data in this review which should be clearly stated in detail under the methods section and a key sentence under the abstract methods section as well.

• Define the first-time used abbreviations i.e. STROBE, PRISMA, and follow the entire manuscript.

• Table 1 research design needs to correct the study design section where qualitative and, Feminist studies are stated in the research design. So consult with the research methods book and state the correct research design of each paper and state it consistently in the entire manuscript.

• Table 2 Check the paper – Orjiako et al 2013 in para 227 and your reference list which is not consistent in the year of the publication. so check all references and strongly suggest using reference management software to make consistency on the citation and reference lists.

• Table 2 Orjiako et al 2013 paper, extracts the consistent data i.e. 95%CI which is missing.

• Table 2 Insert repeat header rows which we can see section heading on each page.

• Para 277 3.6 remove (RoB) but you can define in para 279.

• The discussion section of this manuscript should be very clear, concise and coherent using both empirical and policy sources but para 296 to 316, 320-329 and 341-356 have been written without a single citation/reference. You have not used any single reference.

• Para 4.1 adds strengths in this section, and Para 365 and 377 sections 4.2 & 4.3 should merge under the conclusion section. Recommendations should come after the conclusion of your writing.

• Section 5 conclusion para 383-385 methods stuff should not be repeated in the conclusion section but draw your conclusion and also add present the recommendation section.

• PRASMA flow chart states only those points which have a number and removes those which do not have a number i.e. only (n=0) section.

Reviewers' comments:

Reviewer's Responses to Questions

**Comments to the Author**

1. Is the manuscript technically sound, and do the data support the conclusions?

Reviewer #1: Yes

Reviewer #2: Partly

2. Has the statistical analysis been performed appropriately and rigorously? 

Reviewer #1: Yes

Reviewer #2: N/A

3. Have the authors made all data underlying the findings in their manuscript fully available?

Reviewer #1: Yes

Reviewer #2: Yes

4. Is the manuscript presented in an intelligible fashion and written in standard English?

Reviewer #1: Yes

Reviewer #2: Yes

5. Review Comments to the Author

Reviewer #1: 1. The manuscript is technically sound.However, based on the details of the review, the conclusion part is less reflective of what particularly has been identified.

2. It is already mentioned in the study limitation that the number of studies included in the review is very small, it would have been more insightful if there were more studies included in the review.

3. Eventhough the study follows PRISMA flowchart for selection of the studies and has been shown clearly, the selection of studies and its relevence to the study purpose is of concern. As the title of the study focuses on mental health and help seeking among african migrants, selection of women with post natal depression can possibly reflect mental health conditions and help seeking behavior because of specific factors related to child birth rather than just reflecting the concern as a migrant.

4. The reviewer find the generalizability issue of this study. Very small number of articles has been included with lots of limitations. Authors should make it clear what they are trying to conclude /extract from this study?

Reviewer #2: Thank you for the opportunity to review this systematic review. The review starts off strong with a well written introduction and study objectives. However, the search strategy that the authors have used to find peer-reviewed articles from Africa includes generic geographic search terms (e.g. Africa or Sub-Saharan Africa), but does not include specific African countries (e.g. Ghana, Nigeria). This has potentially led to exclusion of several eligible studies. Secondly, while the four studies the authors have included provide preliminary insights on a few African countries, and a few migrant sub-categories, the limited no. of studies do not provide sufficient findings to answer the research questions. Kindly refer to the following detailed comments for each section:

Abstract

The review objective is clearly stated.

Line 58: Typo in "Results" subsection "two quantitative and two quantitative studies". I suppose the authors meant two quantitative and two qualitative studies.

Line 60-61: "Risk and protective factors associated with mental health and the barriers and facilitators of mental health help-seeking behaviour were identified." - What were these risk & protective factors, and barriers & facilitators? Mention briefly.

Line 64: Conclusion presented in the abstract not justified by results presented in abstract.

Introduction

This section follows a logical and clear argument, is informative and supported by good references. Rationale for selecting the population of interest is well-justified. The research questions are also clear.

Line 73: More recent data on no. of migrants available from IOM World Migration Report 2022/23.

Line 98: Explain the term "affirming practices"

Could the authors clarify or add a sentence on whether the population of interest includes students, labour migrants, migrants moving upon marriage, etc.?

Materials and Methods

Methods section is well-written with good description, particularly of the quality appraisal or risk of bias section.

Line 137-138: Not clear why this line says African migrants to Western countries. The inclusion criteria does not indicate migration to Western countries only.

Line 142: Provide a rationale for selecting the time frame 2012-2023

Search strategy in supplementary file: From the search strategy, I can see that the authors have used "Africa/African" as their search term. This might have led to exclusion of studies that mentioned specific countries such as Nigeria or Ghana rather than Africa. Also including the name of key destination countries could have yielded more results. For example, by using "Sudan" and "UAE" (destination country) in your search strategy, I found: Frontiers | Psychological Distress and Homesickness Among Sudanese Migrants in the United Arab Emirates (frontiersin.org). And by using "Ghana" and "UK" in the search strategy, I found: Perceptions of Ghanaian Migrant Mothers Living in London towards Postnatal Depression during Postnatal Periods There may be many such missed papers. Could the authors please justify this? This is important because the review has included only 4 papers.

Results

Line 276 Suggest replacing the term "non-postnatal depression studies" with the "quantitative study by Orjiako".

The four studies included are quite different from one another in terms of methodology and also the population of interest, because of which there isn't enough data to synthesize or find patterns meaningfully. Similarly, it is not clear which African countries have been included in all the four studies, and therefore it is not easy to ascertain to what extent the various countries in Africa have been represented. African nations are diverse in their culture, economy and socio-political situation. Therefore, the four studies are not adequate in representing the mental health impact on migrants from Africa.

Most importantly, the findings from each study have not been linked to key contextual aspects such as the population, destination country, country of origin and other contextual factors among others. The authors have stated the findings but have not discussed how it links to the migrants culture in their home country, or the culture/circumstances in the destination country.

Because of limited studies, many migrant sub-categories are not well represented including student groups/labour migrants, married vs. unmarried migrants, younger vs. older migrants, etc. While the authors have presented gender disaggregated sample, they have not discussed whether or how mental health impacts/help-seeking vary among men and women.

Discussion

Line 303 "…fear of mental health stigma within certain cultural backgrounds."- What kind of cultural backgrounds? These need to be discussed.

Line 338 "relational factors"- It is not clear which relationships does relational factors include. The authors have specified health professionals, but what about partners, neighbours, colleagues, other family members, etc. Are they included in "relational factors"?

6. PLOS authors have the option to publish the peer review history of their article (what does this mean?). If published, this will include your full peer review and any attached files.

Reviewer #1: No

Reviewer #2: **Yes: **Shraddha Manandhar

---

## [Author Response · Author response to Decision Letter 0]

3 Jan 2024

We thank the Editor and Reviewers for their time and insightful comments. All comments have been addressed in the attached Responses to Reviewers document.

---

## [Editor Report · Decision Letter 1]

29 Jan 2024

Mental health and mental health help-seeking behaviors among first-generation voluntary African migrants: A systematic review

PONE-D-23-19022R1

Dear Dr. Botchway-Commey,

We sincerely appreciate your careful examination and thoughtful analysis of our reviewers' feedback, which appears to fully address each point raised. We’re pleased to inform you that your manuscript has been judged scientifically suitable for publication and will be formally accepted for publication once it meets all outstanding technical requirements.

Kind regards,

Sharada P Wasti, PGDHCM, MHCM, MSc, PhD 

Academic Editor

PLOS ONE

Additional Editor Comments (optional):

One minor suggestion for your final submission: please organize your abstract according to the PLOS One journal format: Purpose, Methods, Results and Conclusion.

---

## [Editor Report · Acceptance letter]

6 Mar 2024

PONE-D-23-19022R1 

PLOS ONE

Dear Dr. Botchway-Commey, 

I'm pleased to inform you that your manuscript has been deemed suitable for publication in PLOS ONE. Congratulations! Your manuscript is now being handed over to our production team.

Kind regards, 

on behalf of

Dr. Sharada P Wasti 

Academic Editor

PLOS ONE